# Collagen-Based Hydrogels Composites from Hide Waste to Produce Smart Fertilizers

**DOI:** 10.3390/ma13194396

**Published:** 2020-10-01

**Authors:** Daniela Simina Stefan, Gabriel Zainescu, Ana-Maria Manea-Saghin, Irene-Eva Triantaphyllidou, Ioanna Tzoumani, Triantafyllos I. Tatoulis, George T. Syriopoulos, Aurelia Meghea

**Affiliations:** 1Department of Analytical Chemistry and Environmental Engineering, Faculty of Applied Chemistry and Materials Science, University Politehnica of Bucharest, 1-7 Polizu str., RO-011061 Bucharest, Romania; simina.stefan@upb.ro; 2National R&D Research Institute for Textile and Leather Division: Leather and Footwear Research Institute, 93 Ion Minulescu str., RO-031215 Bucharest, Romania; gabriel.zainescu@gmail.com; 3Research Center for Environmental Protection and Eco-Friendly Technologies, Faculty of Applied Chemistry and Materials Science, University Politehnica of Bucharest, 1–7 Polizu str., RO-011061 Bucharest, Romania; ana_maria.manea@upb.ro; 4Department of Chemical Engineering, University of Patras, Rio, GR-26504 Patras, Greece; teva@upatras.gr; 5Foundation for Research and Technology-Hellas (FORTH)/Institute of Chemical Engineering Sciences (ICE-HT), Stadiou Str., Platani, GR-26504 Patras, Greece; tzoumani@upnet.gr; 6Department of Chemistry, University of Patras, GR-26504 Patras, Greece; 7Sirmet SA, Engineering and Management, 4-6 Filopoimenos str., 26221 Patras, Greece; ttatoulis@upatras.gr (T.I.T.); gsyriopoulos@sirmet.gr (G.T.S.)

**Keywords:** hide waste, collagen hydrolysate, process optimization, functionalized hydrogels, smart fertilizers

## Abstract

The study aims at reusing and recycling the protein hide waste from the leather industry in ecological conditions by elaborating an innovative procedure in order to obtain a collagen matrix functionalized with nitrogen, phosphorus, and potassium (NPK) nutrients to be used for preparing smart fertilizers. This is an interdisciplinary approach, as it starts from hide waste raw material as a critical industrial waste, which is then subjected to several technological steps by selection of optimal processing parameters, followed by product fabrication at the laboratory, and next scales to the industrial pilot plant to obtain novel agro-hydrogels. In this context, the technology scheme for collagen hydrolysate with encapsulated nutrients was proposed and the process parameters were optimized by functionalization of agro-hydrogels with various natural and synthetic polymers, such as polyacrylamide, poly(sodium 4-styrenesulfonate-co-glycidyl methacrylate) copolymer, starch or dolomite. Based on the laboratory experiments, a pilot plant was constructed and tested. Taking as reference the collagen hydrolysate with encapsulated nutrients, the new fertilizers were adequately characterized by chemical analysis, determination of biodegradability and the degree of release of oxidable compounds in water. Based on the biodegradation mechanism and kinetic analysis of oxidable compounds release, adequate arguments are evidenced to demonstrate that these fertilizers can be applied for amendment of poor agricultural soils.

## 1. Introduction

At present the leather industry faces very high costs to treat and eliminate waste in order to comply with environmental requirements. In this context the strategy is to treat the organic waste (protein and cellulose) by chemical and biochemical processes with the final goal of recycling industrial and agricultural by-products [1]. Studies on leather recycling are focused mainly on obtaining protein composites by biochemical treatments, resulting in protein hydrolysates and protein binders for different applications. Indeed, leather waste is a valuable protein source for diverse areas, such as the automotive industry, animal growth, pharmaceuticals, cosmetics, etc. At the same time, these organic biopolymers are valuable raw materials for agriculture, since their protein waste matrix provides useful elements to ameliorate the composition of poor and degraded soils, and many plants can benefit from elements found in the recycled leather such as nitrogen, magnesium, potassium, etc. [2,3].

The innovative concept of “circular economy” stipulates maintaining the value of products, materials and resources for as long as possible, while minimizing the generation of waste by recycling and reusing them [4]. The leather industry generates large amounts of waste, meaning more than 70% of raw materials (hides and skin), of which almost 99% are currently stored in a landfill [5]. The majority of the waste generated is untreated solid waste, while around 500–600 kg per ton of hide waste contains gelatin, which can be exploited as biofertilizer for agriculture [6]. 

A way to valorize the untreated hide and skin waste is by producing three-dimensional molecular networks named, hydrogels, by cross-linking the proteins hydrolyzed with polymers based on polyacrylamide, polyvinyl alcohol, oligo oxyethylene methacrylate, acrylic acid, maleic acid, cellulose, starch, and gum, that form three-dimensional molecular networks. The hydrogels enriched with nutrients C, N, P, and K can be used as amendments in agriculture for degraded soils [7,8].

Multicomponent absorbent hydrogel-type networks are next generation materials with distinct three-dimensional structure and high swelling capacity. The applications of these materials are diversifying, in recent years entering the fields of agriculture, food, pharmaceuticals, electrical devices and electronics, environmental protection and biomaterials [9].

The hydrolyzed collagen represents a high solubility product that could be used in the food industry as a food supplement or in the cosmetic industry for skin care products [10,11]. By functionalization of collagen hydrolysate with encapsulated nutrients a product with application as a biofertilizer could be obtained [12]. Collagen hydrolysate added to synthetic polymers could improve the biodegradability of plastic materials [13], and collagen-based matrices could also be successfully cross-linked in order to modify mechanical properties or biodegradation rates [14]. 

Collagen could be processed as films, membranes, sponges or skin grafts [15], and collagen hydrogels could be obtained by the hydrolysis of pelt waste, since leather processed in tanneries results in about half pelt waste from a ton of raw hides [16]. Hydrogels exhibit a significant property of swelling and could contribute to a gradual release of nutrients [17]. Collagen hydrogels properties and stability could be enhanced by adding formaldehyde, glutaraldehyde, carbodiimides, polyepoxy compounds, acyl azides and hexamethylene diisocyanate [15] or cross-linked polymers such as polyacrylamide [16], while encapsulating nutrients in the collagen matrix could lead to find potential applications in horticulture [18].

Freshly prepared hide contains almost 33% protein, of which 96.5% are fibrous proteins, and out of these 1% is represented by elastin, 1% is keratin and approximately 98% is collagen [19]. On one side, as the leather industry produces a large amount of solid waste [20], collagen could be recovered from this waste and used in various applications due to its high content of nitrogen [21]. On the other side, collagen obtained from wet blue leather waste after chromium extraction, for example, could be enriched with the mineral potassium (K) and phosphorus (P) for applications as N_collagen_PK fertilizers for the growth of rice plants [22,23].

Collagen hydrogel in the form of gelatin belongs to the class of so-called smart polymer materials that contain functional groups able to modify their volume or other properties in response to environmental stimuli. Hydrogels are cross-linked three-dimensional hydrophilic polymer networks that swell, while they do not dissolve in contact with water. They have the capacity to keep a significant amount of water, which confers to them a soft consistency and low interfacial tension. Their properties depend on the degree of cross-linking, the chemical structure of polymeric chains and the interaction between the network and the surrounding solutions [24]. This high water retention capacity of hydrogels is attributed to hydrophilic groups, such as carboxylic acids, alcohols, and amines, which are largely present in the collagen structure [25].

One of the key points in the recovery of collagen and its derivatives from solid tanned waste is represented by isolation of gelatin to be used as a natural microencapsulation agent in the production of active materials with functional properties. Gelatin was among the first shell-forming materials used in micro-encapsulation and, today, it is still a promising material for creating natural and biodegradable microcapsules, having a wide range of properties such as gel strength, film-forming ability and emulsifying properties. These properties depend largely on the extraction conditions (temperature, pH, time, etc.). It is therefore necessary to optimize the conversion process of hide waste into gelatin to obtain specific properties suitable for materials with high added value. 

The present work aims at reusing and recycling the protein waste from the leather industry under ecologic conditions by elaborating on an innovative process for obtaining collagen composites functionalized with NPK nutrients to be used as fertilizers for amendment of poor soil. Based on mechanisms of chemical modification of the structure of collagenous materials, the optimal process parameters of the pilot plant have been obtained after adequate functionalization with several synthetic and natural polymers. The new biocomposite fertilizers are characterized by chemical and biochemical techniques and collagen hydrolysates functionalized with starch and dolomite; they are considered as promising smart agro-hydrogels for amendment of poor soils.

## 2. Materials and Methods

### 2.1. Materials

Multipolymeric collagen based agro-hydrogels have been prepared using the limed hide waste (no haired) from fleshing and trimming bovine hides (lime fleshing) as the raw material, provided by SC PIELOREX tannery, Jilava, Ilfov county, Romania, (Figure 1). 

Table 1 below lists the main components of the raw hide waste used, including the percentage content of proteins, water, fat and mineral substances. The highest content in proteins justifies the interest for recovery and recycling this hide waste.

A detailed physical–chemical analysis of raw hide waste is presented in Table 2.

The reagents used for this study, such us: H_2_SO_4_, dipotassium phosphate (K_2_HPO_4_), N,N’-methylene bis-acrylamide, starch, poly-acrylamide, dolomite, poly(sodium 4-styrenesulfonate-co-glycidyl methacrylate), H_3_BO_3,_ the homopolymers poly(sodium 4-styrenesulfonate) (PSSNa), were purchased from Sigma-Aldrich (Sigma-Aldrich Chemie GmbH, Taufkirchen, Germany) and used as received. In the pilot-scale experiments industrial salt and water were used.

### 2.2. Preparation of NPK Collagen Hydrolysate and its Functionalization

Gelatin hide is subjected to acid hydrolysis in the presence of potassium phosphate, and the protein hydrolysate is functionalized with monomers or polymers, such as polyvinyl alcohol, acrylamide, maleic anhydride, cellulose, starch, dolomite, etc., in order to obtain smart fertilizers. 

The laboratory experiments for obtaining compounded NPK collagen hydrolysates and their functionalized products have been performed at National R&D Institute for Textile and Leather–Division Leather and Footwear Research Institute, Bucharest, Romania, (Figure 2).

The technology diagram for obtaining collagen hydrogel products is illustrated in Figure 3, and it can be briefly described by the following steps:obtaining the collagenous matrix by acid hydrolysis of the gelatine hide at 86–90 °C, with 2.7–3.6% H_2_SO_4_ for 1.5–3.5 h;additivation of collagen hydrolysate with PK nutrients by treating with a K_2_HPO_4_ solution of 10–20% concentration;preparing in parallel a polymeric solution of starch or polyacrylamide gel by continuous mixing, dezaeration and addition of N,N’-methylene bis-acrilamide as reticulant agent;functionalization of NPK collagen hydrolysate by copolymerization with starch, poly-acrilamide, or dolomite suspension;obtaining smart collagen-based fertilizers by encapsulation in natural organic oils.

The experiments are explained in detail by the following four examples:


**Example 1.**
*An amount of 6500 g limed hide waste is weighed and subjected to treatment with powder lime in proportion of 6–10% within a plastic basin for 3–6 days; then an acid hydrolysis is carried out in the presence of 2.7–3.6% H_2_SO_4_, in a 50 L autoclave with a double jacket and shaker, at 80–98 °C, for 1.5–2.5 h. A solution of 1.8–3.5% dipotassium phosphate (K_2_HPO_4._3H_2_O) and 24–30% corn starch is added; methylen bis-acrylamide (0.5 g dissolved in 50 mL water) is used as a chemical reticulant agent, resulting in a copolymer of collagen chemically grafted with starch. The hydrolysis continues for 1.5–2 h, at 78–85 °C temperature, resulting in a collagen-starch agro-hydrogel, denoted as AMI, a superabsorbant, in the form of a consistent white, gelatinous, elastic paste, with pH 6.7–7.3, which is dryed as foils and delivered in plastic bags (Figure 4a). If instead of starch, a solution of 18–25% polyacrilamide gel is used, another agro-collagen hydrogel was obtained, denoted as POLY.*



**Example 2.**
*An amount of 3800 g gelatin hide waste is weighed and mixed with 3–4.5 L industrial water at 25–35 °C temperature containing 2–3.5% H_2_SO_4_, and 0.5–1% industrial salt (NaCl); the mixture is let to react in a 50 L autoclave with double jacket and shaker, at 80–98 °C, for 1.5–2.5 h; then 15–23% natural dolomite is added. A solution of 2.5–3.5% dipotasium phosphate (K_2_HPO_4_.3H_2_O) and 0.05–0.1% boric acid (H_3_BO_3_) and the cold product is extruded, resulting in a collagen–dolomite agro-hydrogel, denoted as DO, in the form of 4–6 cm bars of greenish color, with pH 5.8–6.8, (Figure 4b).*


In order to optimize the collagen hydrolysate encapsulation, reactive copolymers with various epoxide groups content were also synthesized and tested. Thus a thorough research in terms of optimization of the synthesis of the water soluble copolymer poly(sodium 4-styrenesulfonate-co-glycidyl methacrylate) (P(SSNa-co-GMAx)) was previously conducted, where x = 2, 5, 20, 30, and 40% mole GMA [12].


**Example 3.**
*The copolymer P(SSNa-co-GMA20) was selected as having the appropriate characteristics to control the behavior of the hydrogels that will be used as fertilizers. Specifically, skin pelt waste was acid hydrolysed with 1–1.6% sulfuric acid at 85–90 °C. The hydrolysate was further enriched by adding 7–14% dipotassium phosphate. At the enriched collagen hydrosylate, the copolymer P(SSNa-co-GMA20) was added at a final concentration of 0.5% and left overnight to react at 80 °C. On the next day, the solution was cooled down and after formation of two phases the supernatant solution was removed and the remaining gel, which was actually the biopolymer, was extruded. The biopolymer, denoted as PSSG, was left to dry at room temperature and the final product with high viscosity was solidified in time.*


In this context, a variant of the technology process has been developed by FORTH-ICHT Group, Patras, Greece by using this synthetic copolymer for collagen hydrolysate functionalization, P(SSNa-co-GMAx), while the corresponding tests have been carried out on the industrial pilot plant built at SIRMET S.A., Patras, Greece, (Figure 5).


**Example 4.**
*In an alternative procedure, the collagen hydrosylate reacted with the copolymer P(SSNa-co-GMA20) in a pilot-scale reactor, having a different collagen hydrolysate/copolymer ratio. This was done in order to further investigate the release characteristics. Specifically, 1.6 kg of collagen hydrolysate was added in the reactor of the pilot plant and dissolved in 23 L of water, resulting in a 7% solution of collagen hydrolysate. When the dissolution was completed, 400 g of copolymer P(SSNa-co-GMA20) was added in three portions and left overnight to react at 80 °C. When the mixture reached room temperature and two phases were formed, the supernatant was removed and the remaining gel, the biopolymer, was extruded. The product was left for drying at room temperature until solidification. The experiments regarding this biopolymer are in development and the results will be the topic of a future publication.*


### 2.3. Characterisation of Compounded Hydrogels

Physical chemical characterization of smart agro-hydrogels is given in Table 3, where for the studied samples the following codes have been allocated: HC—collagen hydrolysate, Ref-HC—collagen hydrolysate with nutrients encapsulated as reference sample, PSSG—Ref-HC functionalized with P(SSNa-co-GMAx) copolymer, POLY—Ref-HC functionalized with poly-acrylamide, AMI—Ref-HC functionalized with starch, DO—Ref-HC functionalized with dolomite.

One may observe that the highest organic content is found in the AMI fertilizer functionalized with the natural polymer starch, while the lowest organic content corresponds to the sample where the collagen is functionalized with the synthetic polymer, PSSG. Moreover, in all samples doped with nutrients, the nitrogen content ranges between 8 and 10.5%, in phosphorous between 5 and 7.7%, and for potassium between 8.2 and 10.6%, thus confirming the encapsulation efficiency of this collagen gelatin matrix.

### 2.4. Determination of Biodegradability

For these new bio-fertilizers with a polymeric structure, the most appropriate methods for biodegradability determination were considered according to the standard methods used for similar polymeric waste: SR EN ISO 14852/2005 – Determination of the final aerobic biodegradability in aqueous medium of plastic materials, by analyzing the evolved carbon dioxide;SR EN ISO 14855-1/2008 - Determination of the final aerobic biodegradability in composting controlled conditions by measuring the quantity of evolved carbon dioxide.

**Method**. The laboratory device consists of the following components: an air pump, a vessel with a solution 40% KOH for removal of CO_2_ from air, a glass column filled with silicagel for air drying, a biodegradation vessel containing the nutrient solution, an inoculum, a known amount of fertilizer, a vapor condensing vessel, and a vessel containing a solution of 0.1 N NaOH for CO_2_ collection. Aliquots of this solution were daily sampled and titred in two steps with a solution of 0.1 N HCl for determination of the unreacted NaOH and of the bicarbonates resulting from the process.

For determination of biodegradability in composting conditions, a similar installation was used, with the only difference being that the biodegradation vessel contains compost instead of nutrient solution.

**Release degree of oxidable compounds in water.** The release degree of oxidable compounds (organic and inorganic) in the presence of KMnO_4_ (CODMn) of all bio-fertilizers tested reflects their stepwise decomposition in aqueous medium in dynamic conditions, according to the method described in a previous paper [12]. 

## 3. Results and Discussion

### 3.1. Ecological and Mechanistic Approaches on Smart Biofertilizers

The complex issues related to reuse of protein waste from the leather industry are focused on the determination of the chemical composition of hide waste in tight connection with various possibilities for their recovery and recycling by using emerging biotechnology innovations. All of them having as the final goal the substantial abatement of environmental pollution according to advanced concepts of sustainable development.

By chemical hydrolysis at temperatures over 100 °C of hide waste from fleshing and trimming pelts various polypeptide fractions until amino acid level can be obtained [26].

There are two main areas, namely agriculture and medicine, where controlled release has extensive applicability. Both of these areas need the use of some biocompatible, nontoxic matrices, which are biologically inert and biodegradable. Within tanneries, only around one-third of the total raw material (animal hides) is transformed into fine leather, while the remaining two thirds is mostly a protein source, which can be reused as agrocollagen fertilizer in most leather processing countries.

Composite materials based on collagen hydrolysate prepared from gelatin pelts waste are known as nontoxic products [26], since they are produced based on a similar technology used for obtaining medical collagen for cosmetic applications, as is explained in the scheme below (Figure 6). The main difference consists of the fact that for medical collagen for hydrolysis the entire gelatin pelt is used, while for agro-collagen the gelatin pelt waste is used after cutting the gelatin pelt for fabrication of footwear and leather goods by tanning. 

Until recently, the collagen hydrolysate was largely used as a nutritive protein additive for young animals feeding in Italy, the Czech Republic and Russia, but this has been forbidden in Europe after the “mad cow disease”.

Another difference between medical collagen and agrocollagen fertilizers is in the drying step. While for medical collagen the drying is allowed by liofilization or atomization (frequently used for medicines, cosmetics, implantology, etc.), in the case of agrocolagen these procedures are not applicable, nor maintainable in a liquid state, since the proteins are easily degraded in air, entering in putrefaction (bad smelling), neither could the other stabilizing agents be used, as they are toxic.

As shown in the ecological balance from Figure 7, in the leather tanneries industry the energy consumption is in range of 15–20 GJ per ton of raw hide. During processing of raw animal skins, the environment is affected by air emissions, wastewater and solid waste. The air emissions consist of CO, CO_2_, and NO_2_ from heating the hide with gas, repulsive odor caused by H_2_S and NH_3_, resulting from unhairing/liming, deliming and degradation of protein components of the waste_,_ solvents and formaldehyde from the finishing processes [27].

Wastewater in an amount of cca 45 m^3^/tonne of raw hides contain salts, especially chlorides, fats, proteins, preservatives (soaking), lime and ammonium salts, ammonia and sulphides (fleshing, trimming, and bating), chromium salts, and polyphenolic compounds (tanning); dyes and solvents (wet-finishing), and around 450 kg sludge. Each tonne of raw hides results in 200–250 kg leather, while untanned solid waste is around 250 kg from the operations of slaughterhouses (trimming, shaving, and fleshing); the total solid waste that is produced is about 450–600 kg [5,28].

For ecological reasons other solutions for reuse and recycling of pelt waste from tannery industry have been proposed, one of them being for collagen hydrolysate with micronutrients incorporated to be used as fertilizer for rehabilitation of poor soils [29,30,31].

In this context, hydrogels can be obtained by two major mechanisms: hydrogels with covalent or irreversible links and hydrogels with reversible or physical links. The second category includes various subclasses such as ionic interactions (ionic hydrogels or cross-linked polyelectrolyte complexes) and secondary interactions (“entangled” hydrogels, grafted or complexed hydrogels, etc.), [32].

Collagen, the main component of hide and hide waste, is considered a high-tech bio-molecule (currently, collagenous tissue can hardly be replicated by synthetic means, due to its three-dimensional crosslinks) and possesses functional groups (-CO–NH-, -OH, -NH_2_, and -COOH), with very good reactivity and cross-linking properties. Performance of a natural polymer can be improved by using the strategy generally known as **“**chemical modification**”** [1]. 

Similar to other polymers, collagenous materials could be subjected to the following types of chemical modifications:
Polymer-analogous transformations, when only the functional groups belonging to amino-acids residues are involved;Grafting, characterized by linking at collagen macromolecule of oligomer type structures (${\overset{-}{M}}_{o}$ < 10,000 Da);Reticulation, devoted to transformation of linear and branching configurations in tridimensional configurations.

Considering chemical modification of collagenous materials only, analogous-polymer transformations will be taken into account, which consist mainly in treating with micro- or macromolecular reagents, capable of interacting with one or several functional groups attached to the polypeptidic backbone, according to the following reactions:

1. Direct transformation:
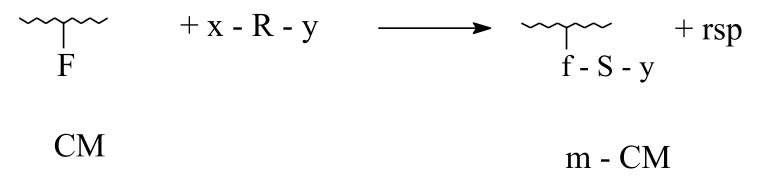



2. Indirect transformation:

-with noncatenable coupling agents (NCA) 
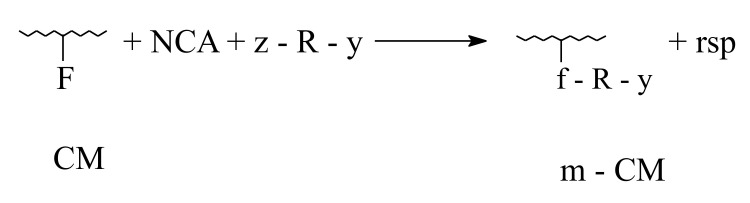


-with catenable coupling agents (CCA) 
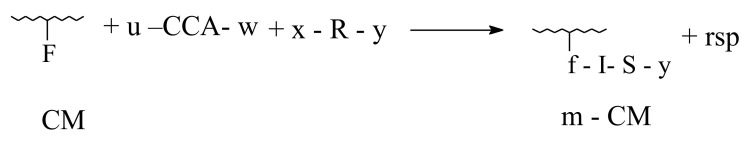


With reference to the technology scheme presented in the Figure 3, the composite material obtained by functionalization with starch occurs by chemical binding with collagenous functional groups (-CO-NH-, -OH, -NH_2_ and -COOH). Moreover, due to the strong intermolecular hydrogen bonds and high polarity discrepancies between collagen hydrolysate and starch, N,N’-methylene bis-acrylamide should be added as a reticulant agent. In this way a valuable copolymer of starch chemically grafted with collagen hydrolysate is obtained, with a high capacity to incorporate micronutrients and to control their release into the soil and plants. Such biocomposites endowed with efficient released control of some active compounds important both in medicine or in agriculture are recently termed as smart biopolymers, as they mimic intelligent processes in living beings.

### 3.2. Biodegradability of Fertilizers in Aqueous or Composting Environments 

This test was performed according to the standard SR EN ISO 14852/2005 for biodegradability degree of plastic materials in water in aerobic conditions, and it is based on the determination of carbon dioxide release, monitored for 75 days. 

As referring to the release rate, four regions of different lengths depending on the fertilizer type could be identified, as is illustrated in Figure 8 and Figure 9:The stagnant zone corresponds to the first step, where the biodegradation process is initiated and the biodegradation rate is small (0–6 days);The acceleration zone, when biodegradation degree is linearly increasing in time (2–50 days);The slowing zone, where even though there is an increasing tendency, the biodegradation rate is smaller (14–56 days);The stationary zone, where the biodegradation degree reaches its maximum value and the process rate tends to zero (41–75 days).

For three selected fertilisers the biodegradability degree has been also tested in aerobic composting conditions, as described in the Section 2.4, in order to simulate real agriculture environment (Figure 9).

Table 4 presents the polynomial regression equations for the biodegradability degree in time for fertilizers tested in water and in composting conditions.

A comparison between the evolution in time of the biodegradation degree in water and in composting conditions for the fertilizers tested is presented in Table 5.

The evolution in time of the biodegradability degree in water (W) indicates the maximum value of 99% for collagen hydrolysate HC, as expected, while for fertilizers with NPK nutrients encapsulated the following values have been obtained: 74% for Ref-HC, 80% for AMI, and over 60% for POLY and PSSG products. These results are in perfect agreement with the fertilizer structures, being higher for the products with biopolymer content and smaller in the case of compounds functionalized with synthetic polymers (polyacrylamide and P(SSNa-co-GMAx) copolymer). Moreover, one may also notice that the biodegradability degree is lower in composting conditions than in water, but the composting conditions are nearer to a real environment when applying the nutrients on soil rather than in water. The best results among agro-gels are obtained for collagen hydrolysate with nutrients encapsulated and functionalized with starch (AMI), which will be further tested in more detail in a real environment on cultivated soils.

As referring to the release mechanism one may suppose that in the first stage, the organic and inorganic compounds with high solubility are removed, such as: peptides with short chains, soluble polymers (starch), amino acids, glucides (mono- and di-saccharides), nitrate salts, acids, etc. In the second stage the rate of release of oxidable compounds decreases for all fertilizers; this can be assessed to organic compounds with low solubility, such as proteins, collagen, and degradable polymers [12].

### 3.3. Leaching the Oxidable Compounds in Laboratory Dynamic Systems

Release of oxidable compounds (organic and inorganic) in water from the tested agro-hydrogels has been characterized by their stepwise decomposition in aqueous medium in dynamic conditions, monitored by determination of chemical oxygen demand (CODMn) [12].

Leaching degree of oxidable compounds from tested fertilizers during one month is shown in Figure 10.

It is obvious that collagen hydrolysate (HC) is practically completely degraded during chemical oxidation (>99%), while the collagen functionalized with starch (AMI) has the highest leaching degree among all fertilizers tested (90%). The lowest values are registered for agro-hydrogels functionalized with synthetic polymers, polyacrylamide (POLY, 75%) and P(SSNa-co-GMAx) copolymer (PSSG, 73%). Moreover, if we consider the drug release over one unit of time (one day) we are able to evaluate initial kinetics (burst release), which is maximum, as expected, for initial collagen hydrolysate, followed at half by the NPK doped hydrogel compounded with starch biopolymer, and much smaller for the hydrogels functionalized with synthetic polymers, which confer a more stable and rigid surface configuration.

In order to assess the release mechanism, the experimental data have been processed according to the empirical Rigger-Peppas equation [33,34]:R_f_ = k t^n^(1)
where R_f_ is the released fraction, k is a kinetic constant depending on the nature of the material, and n is the diffusional exponent, depending on type of transport, hydrogel geometry and polymer polydispersity. Usually n = 0.5 when the release follows the Fick model diffusion, and n = 1 when surface deterioration dominates the release. In most cases n is between 0.5 and 1, meaning that several mechanisms control the release such as: diffusion, electrostatic interactions, hydrophobic associations, cleavable covalent linkage, and degradation.

These kinetic parameters calculated for the curves from Figure 10 are given in Table 6 for the entire time period analyzed of 26 days.

One may notice that all the values for the exponent n are above 0.5, what indicates a more complex release mechanism, and not a Fickian diffusion control for all the samples tested. Indeed, a more detailed analysis was carried out for the kinetic parameters calculated using the logarithmic form of the above equation, when two linear regions were identified on limited time intervals; the corresponding kinetic parameters are collected and presented in Table 7.

A careful analysis of the data from Table 7 reveals at least two transport mechanisms for the release of oxidable compounds: (1) the initial step during the first 3–10 days, when the kinetic exponent, n > 1, and (2) the second step when this parameter takes values around 0.5. These results can be interpreted by the strong chemical attack initially caused by the oxidation agent (KMnO_4_), which heavily destroys the hydrogel matrix. This process is more rapid in the case of initial noncompounded collagen hydrogel, HC, and of the fertilizer obtained by functionalization with starch biopolymer, AMI, which is in perfect agreement with their higher burst release. During the second step the oxidation products are released after a diffusion control mechanism. Once again it is confirmed that the biofertilizer obtained by functionalization of collagen hydrogel with starch is easily degradable, while it exhibits the longest release period, making it a very promising material for poor soil amelioration. 

## 4. Conclusions

This study is focused on recycling hide waste resulting from the leather industry by capitalising their valuable protein components to obtain efficient collagen-based hydrolysates as smart fertilizers for poor soils amelioration. In this aim, a series of measurements has been carried out, on the design and production of various fertilizers by functionalization of collagen hydrolysates encapsulated with P and K nutrients with some synthetic and natural polymers, including polyacrylamide, copolymer poly(sodium 4-styrenesulfonate-co-glycidyl methacrylate), and starch as a biopolymer. These agro-hydrogels have been characterized for their physical chemical properties and some preparatory agrochemical tests have been developed, including biodegradability and leaching oxidable compounds in water. It was demonstrated that the most promising fertiliser is the collagen hydrolysate functionalised with starch; it had the best biodegradability and degree of leaching of active compounds toward water and soils.

## Figures and Tables

**Figure 1 materials-13-04396-f001:**
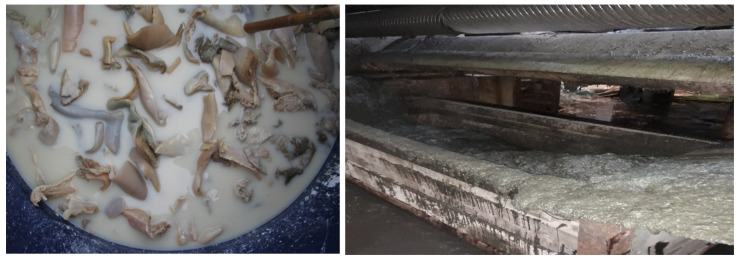
Limed hide waste (gelatin and lime fleshing) at SC PIELOREX Jilava, Romania.

**Figure 2 materials-13-04396-f002:**
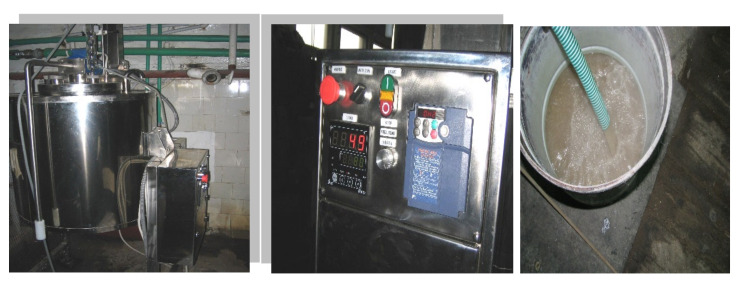
Laboratory plant for obtaining SMART agro-collagen and its functionalization products.

**Figure 3 materials-13-04396-f003:**
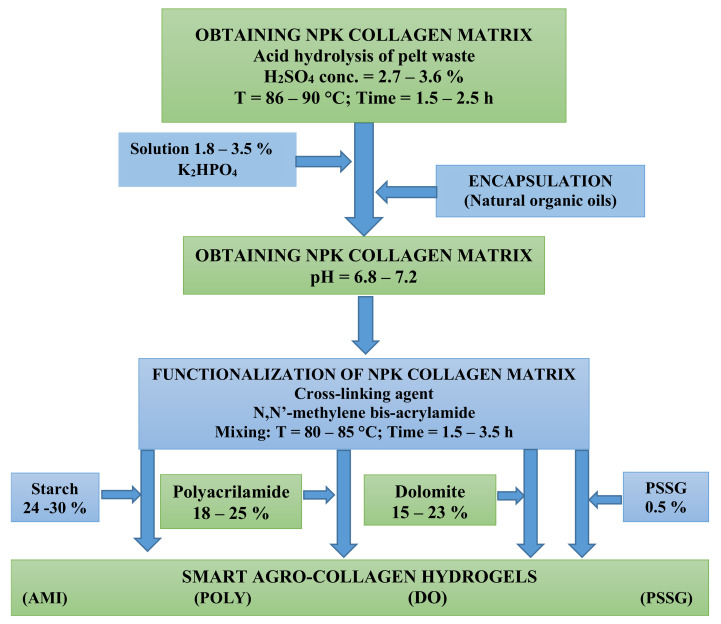
Technology scheme for obtaining smart agro-collagen hydrogels.

**Figure 4 materials-13-04396-f004:**
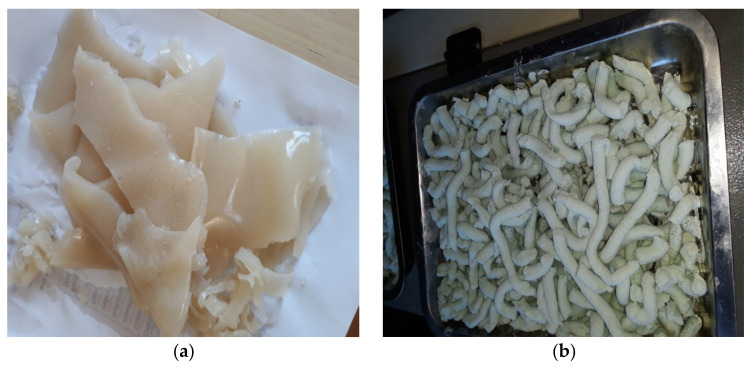
Agro-hydrogels compounded with (**a**) starch (AMI) and (**b**) dolomite (DO)

**Figure 5 materials-13-04396-f005:**
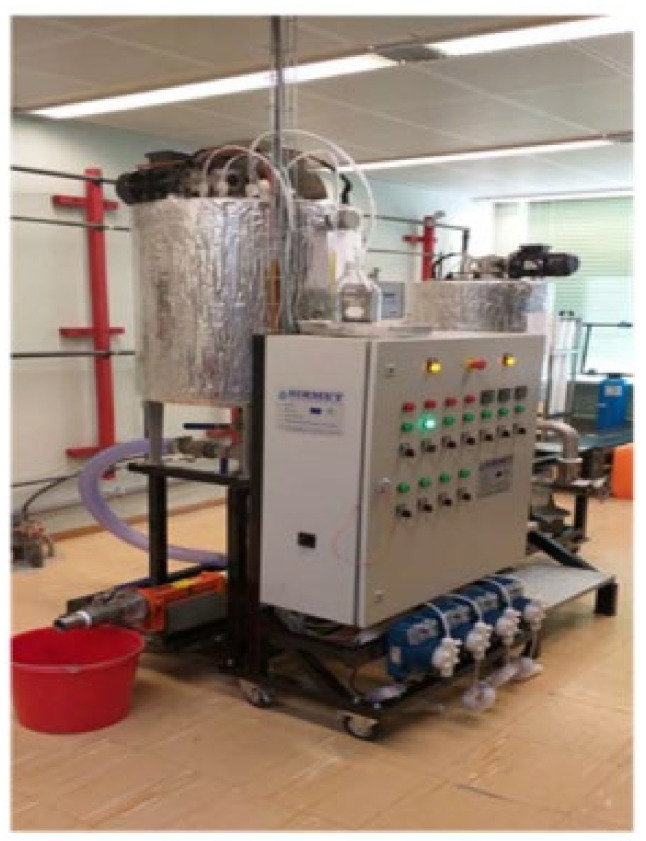
The pilot plant built at SIRMET, Patras.

**Figure 6 materials-13-04396-f006:**
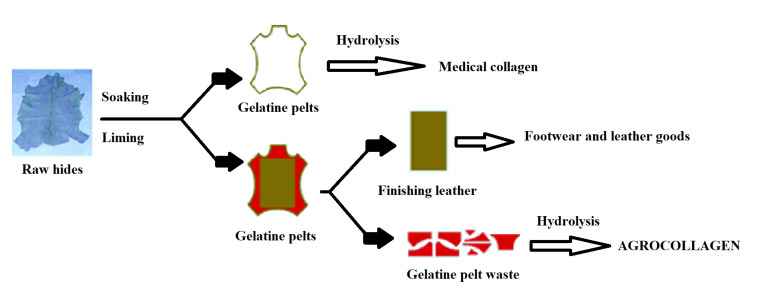
The comparison between the production of medical collagen and agro-collagen.

**Figure 7 materials-13-04396-f007:**
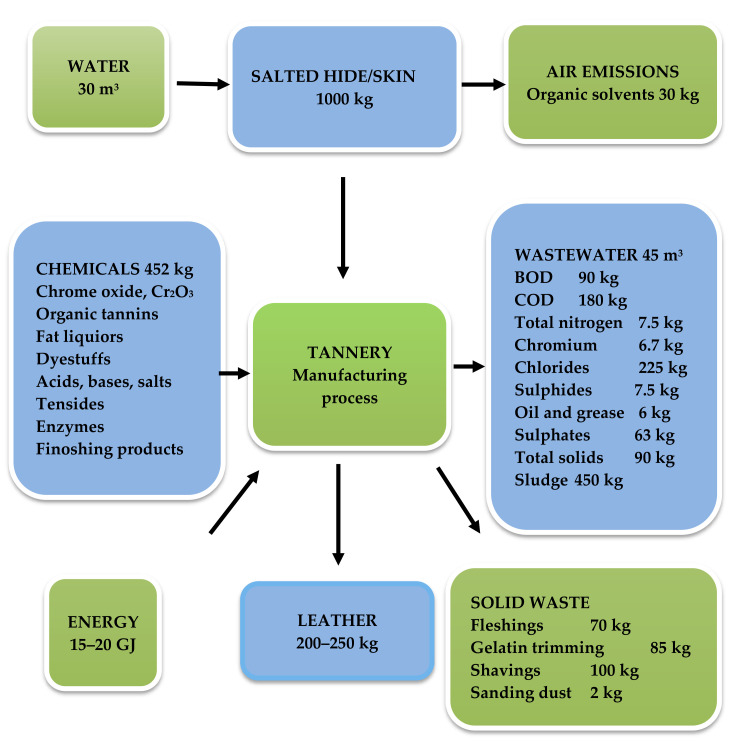
The ecological balance sheet during processing one tonne of raw hides

**Figure 8 materials-13-04396-f008:**
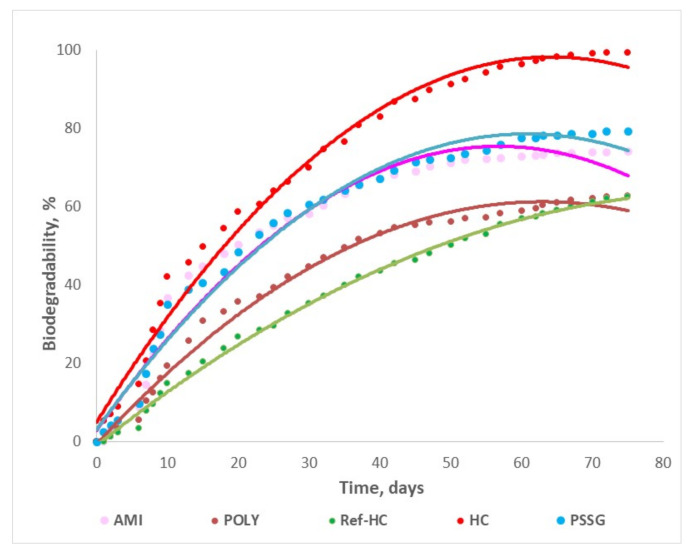
Evolution in time of the biodegradability degree of the tested fertilizers in water in aerobic conditions.

**Figure 9 materials-13-04396-f009:**
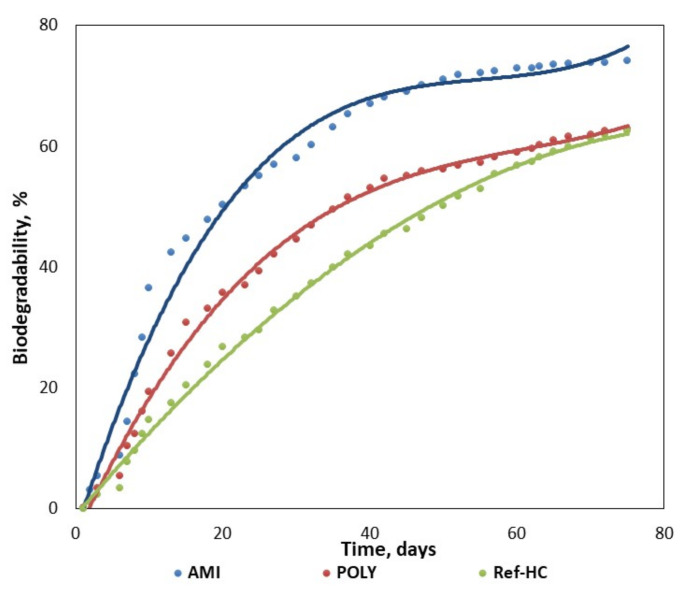
Evolution in time of the biodegradability degree of the tested fertilizers in composting conditions.

**Figure 10 materials-13-04396-f010:**
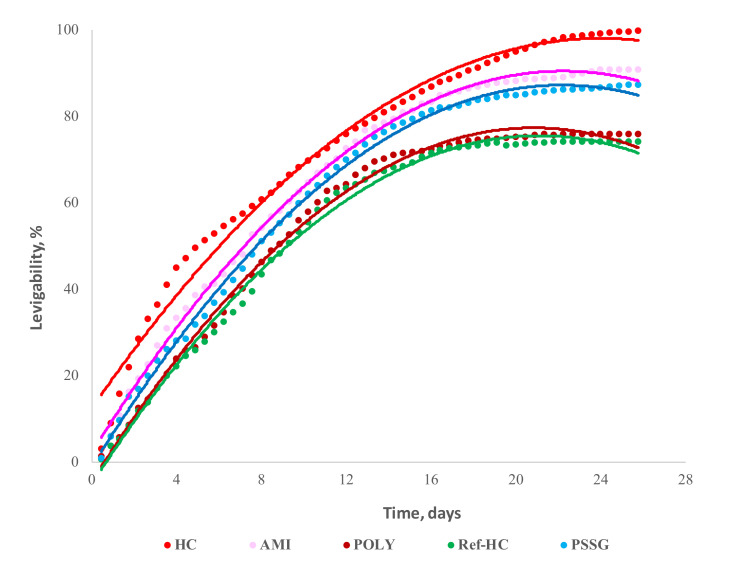
Profile of the release degree of oxidable compounds from the tested collagen-based fertilizers.

**Table 1 materials-13-04396-t001:** Chemical components of raw hide waste.

Component	Hide Fleshing	Trimming Gelatin	Splitting Gelatin	Lime Raw Hide
Proteins, %	10–50 *	21–84 *	22–90 *	23–61 *
Water Content, %	80	75	75	62
Fat Substances, %	7–35 *	1–4 *	0.3–1.2 *	13.5–36 *
Mineral Substances, %	3–15 *	3–12 *	2.2–8.8 *	1–3.0 *

* values are reported as dry substance.

**Table 2 materials-13-04396-t002:** Physical–chemical characteristics of raw hide waste.

Nr.Crt.	Parameter	Values	Standard Methods
1	Volatile mater, %	68.74	SR EN ISO 4684-2006
2	Fat maters, %	2.30	STAS 145/20-1988
3	Total, ash, %	1.18	SR EN ISO 4047-2002
4	Total nitrogen, %	13.76	SR ISO 5397-1996
5	Dermal substance, %	77.33	SR ISO 5397-1996
6	Metal oxides, %	1.01	ICPI Methods
7	pH- aqueous extract	7.34	SR EN ISO 4045-2008

**Table 3 materials-13-04396-t003:** Chemical analysis of the tested fertilizers.

No	Fertilizer Parameter	UM	HC	Ref-HC	PSSG	POLY	AMI	Method of Analysis
**1**	**Dry Substance**	%	24.18	61.22	65.8	21.59	26.38	SR EN ISO 4684: 2006
**2**	**Ash**	%	2.48	22.36	25.18	17.23	16.38	SR EN ISO 4047: 2002
**3**	**Total Nitrogen**	%	10.36	10.55	10.14	12.13	8.29	SR ISO 5397: 1996
**4**	**Soluble Phosphorus, P_2_O_5_**	%	-	7.67	6.75	5.79	5.54	SR EN 15959: 2012
**5**	**Soluble Potassium, K_2_O**	%	-	10.62	8.21	8.40	10.07	SR ISO 5397: 1996
**6**	**Total Organic Carbon, TOC**	%	46.2	45.2	37.56	48.1	64.32	SR EN 13137/2005
**7**	**pH**	units	6.70	7.20	6.87	6.76	6.20	STAS 8619/3-1990

**Table 4 materials-13-04396-t004:** Polynomial regression equations for the biodegradability degree in time for fertilizers, in water (W) and in composting (C) conditions.

Fertiliser Type	Biodegradability, %, in Time Polynomial Regresion Ecuation	R²
HC-W	y = −0.0226x^2^ + 2.903x + 5.0378	0.9872
Ref-HC-W	y = −0.0081x^2^ + 1.4468x − 0.9437	0.9972
Ref-HC-C	y = −0.0082x^2^ + 1.457x − 1.1262	0.997
AMI-W	y = −0.0225x^2^ + 2.5537x + 2.972	0.9672
AMI-C	y = 0.0004x^3^ − 0.0683x^2^ + 3.8722x − 4.0301	0.9844
POLY-W	y = -0.0156x^2^ + 1.9655x − 0.5645	0.9891
POLY-C	y = 0.0002x^3^ − 0.0399x^2^ + 2.6786x − 4.6409	0.9956
PSSG-W	y = −0.0205x^2^ + 2.4853x + 3.1857	0.9811

**Table 5 materials-13-04396-t005:** Evolution of biodegradability degree in water (W) and in composting conditions (C).

No	Biodegradation Degree, % Days	HC	Ref-HC		AMI		POLY		PSSG
		W	W	C	W	C	W	C	W
**1**	**10**	42	35	17	37	33	15	12	20
**2**	**20**	58	48	33	50	48	27	21	35
**3**	**75**	99	74	50	80	64	62	40	63

**Table 6 materials-13-04396-t006:** Kinetic parameters for oxidable compounds released from fertilizers tested on the entire time period.

Fertiliser Type	k	n	R2
HC	0.137	0.6693	0.9159
Ref-HC	0.048	0.9514	0.924
POLY	0.056	0.9059	0.9212
PSSG	0.071	0.8644	0.9097
AMI	0.088	0.7995	0.9212

**Table 7 materials-13-04396-t007:** Kinetic parameters for oxidable compounds released on two time intervals.

Fertiliser Type	Time Interval, Days	k	n	R^2^
HC	0–2.67	0.097	1.2466	0.9838
	2.67–25.78	0.242	0.4527	0.9929
Ref-HC	0–9.78	0.034	1.2524	0.9725
	9.78–25.78	0.338	0.4551	0.8232
POLY	0–10.22	0.041	1.1732	0.9906
	10.22–25.78	0.383	0.4309	0.8333
PSSG	0–5.11	0.048	1.3215	0.9325
	5.11–25.78	0.132	0.6285	0.9373
AMI	0–3.56	0.06	1.3017	0.9707
	3.56–25.78	0.171	0.5497	0.9511

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
