# Peer review of "Collagen-Based Hydrogels Composites from Hide Waste to Produce Smart Fertilizers"

_materials, 2020, doi:10.3390/ma13194396_

Round 1

Reviewer 1 Report

The authors have extensively discussed the industrial applicability of their technique of reusing the leather waste. The concept is well supported by appropriate literature and facts. I have just 2 comments,

  1. Authors should add the important reactions in Figure 3 along with the flowchart.
  2. On page 12, line 351, authors mention dynamic conditions. They should comment on the temperature dependence (range) during the release of these oxidable compounds.

Author Response

Dear Editor,

We express our thanks for positive appreciations of two of reviewers. At the same time we analyzed carefully all the observations and corrections suggested by reviewers and operated the changes and completions as requested on the initial version of the article with new paragraphs added in yellow, thus contributing to better fit the paper content to the journal scope.

Reviewer 1

The authors have extensively discussed the industrial applicability of their technique of reusing the leather waste. The concept is well supported by appropriate literature and facts. I have just 2 comments,

  1. Authors should add the important reactions in Figure 3 along with the flowchart.

As we tried not to complicate too much the scheme from Figure 3, we add a sequence of important reactions in lines 327-371, at Results and discussion, section 3.1. Ecological and Mechanistic Approaches on Smart Biofertilizers.

  1. On page 12, line 351, authors mention dynamic conditions. They should comment on the temperature dependence (range) during the release of these oxidable compounds.

These laboratory experiments have been performed at 20oC, while temperature dependence has been not taken into account in this study. We assumed that this temperature corresponds to average conditions when consider real environment for fertilisers degradation in soil during spring time.

Reviewer 2 Report

Authors describe the production of collagen based hydrogels from waste for their fertilizer application.  In the present form, the article is out of the scope and it could be more suitable for its consideration on Process (MPDI).

Overall, the article is extremely mixed up, being some part hard to follow.

Here you could find some issues (not limited to):

The state of art present in the introduction tries too many subject but very superficial without a deep review existing bibliography on the area. In addition, the definition of hydrogel is not acceptable.

What is NPK? You mentioned in the abstract without any definition.

Materials and methods must be divided, that is, describe the materials (purity, provider,…) used and in a separate section the experimental part. The details of the experiments must be adequately described in order to be reproducible of the readers if they want to, at the present for I have my doubts.

Which are the characteristics of the leather used?

How is the characterization of table 1 carried out?

How is the characterization of table 1 carried out?

Author Response

Dear Editor,

We express our thanks for positive appreciations of two of reviewers. At the same time we analyzed carefully all the observations and corrections suggested by reviewers and operated the changes and completions as requested on the initial version of the article with new paragraphs added in yellow, thus contributing to better fit the paper content to the journal scope.

Authors describe the production of collagen based hydrogels from waste for their fertilizer application.  In the present form, the article is out of the scope and it could be more suitable for its consideration on Process (MPDI).

            The recommendation of reviewer that this paper could be more suitable for its consideration on Process (MPDI) is quite reasonable as our research was intended to cover an integrative approach, starting from leather waste as a critical industrial waste, its processing by several technology steps, with selection of optimal parameters for obtaining some new materials not only at laboratory, but also at pilot scale. However, we considered that this comprehensive approach is very useful in demonstrating the final goal of our research to obtain useful materials to be applied as fertilisers in real agriculture conditions as it will be reported in the next publication. To be more specific to materials and to respond to reviewer observation, we extended the section devoted to characterization of materials by adding 3 new figures (8-10) and 3 tables (4, 6 and 7) on biodegradability and mechanism of oxidable compounds release.

Overall, the article is extremely mixed up, being some part hard to follow.

            The paper has been completely revised, being improved and many aspects have been clarified to be easier to follow.

Here you could find some issues (not limited to):

The state of art present in the introduction tries too many subject but very superficial without a deep review existing bibliography on the area. In addition, the definition of hydrogel is not acceptable.

            The introduction was rewritten, the reference list was completed from 29 to 34, with focus on hydrogel concept, a special paragraph being added before its final part.

What is NPK? You mentioned in the abstract without any definition.

            The capital letters NPK means the chemical symbols of elements: nitrogen, potassium and phosphorous, the main nutrients provided by fertilizers for poor soils amendment. However, to remove any confusion, the term NPK collagen matrix in the abstract and in the text has been replaced by ‘a collagen matrix functionalized with NPK nutrients’.

Materials and methods must be divided, that is, describe the materials (purity, provider,…) used and in a separate section the experimental part. The details of the experiments must be adequately described in order to be reproducible of the readers if they want to, at the present for I have my doubts.

            The Materials section is separated from Preparation and Characterization methods at section 2.1. It is mentioned that the reagents for laboratory tests have been used as received, without further purification; being a research for capitalization of leather waste for application as fertilisers on soil, an advanced or special purity is not need. However, to clarify this aspect, the following sentence has been added: ‘In the experiments at pilot scale industrial salt and water have been used’

Which are the characteristics of the leather used?

            Two new tables (nr 1 and 2) have been added, comprising the content in the main chemical components of the raw material used: proteins, water, fat and mineral substances; detail physical-chemical characteristics are also given.

How is the characterization of table 1 carried out?

            The table 1 now became table 3 and the data from this table have been adequately commented. 

Reviewer 3 Report

The research article presents the use of collagen-based hydrogels manufactured in part with recycled waste materials from leather to create nutrient rich polymer composites suitable for soil fertilizers. The use of previous research on optimization of the hydrogel cocktail was properly utilized as a basis for this work and was appropriately cited. The experimental methods and approach were suitable.

The author gives an excellent discussion of current practices in Europe with historical context, giving credence to the need for this type of technology. 

Discussion:

  • If possible, the understanding of some of the Tables would be easier if they were in graphical form. Specifically, a graph showing the release rates during the 4 different release zones (Table 2). This sort of release mechanism is widely used in drug release of hydrogels. The author may benefit from fitting a sigmoidal or appropriate release kinetics equation to their system. Two articles that may be of use:
    • Lee, P. I. (1985). Kinetics of drug release from hydrogel matrices. Journal of Controlled Release2, 277-288.
    • Li, J., & Mooney, D. J. (2016). Designing hydrogels for controlled drug delivery. Nature reviews. Materials1(12), 16071. https://doi.org/10.1038/natrevmats.2016.71

Grammatical:

  • Line 48:  "row" should be "raw"
  • Line 256: "Liofilization" should be "lyophilization"

Author Response

Dear Editor,

We express our thanks for positive appreciations of two of reviewers. At the same time we analyzed carefully all the observations and corrections suggested by reviewers and operated the changes and completions as requested on the initial version of the article with new paragraphs added in yellow, thus contributing to better fit the paper content to the journal scope.

The research article presents the use of collagen-based hydrogels manufactured in part with recycled waste materials from leather to create nutrient rich polymer composites suitable for soil fertilizers. The use of previous research on optimization of the hydrogel cocktail was properly utilized as a basis for this work and was appropriately cited. The experimental methods and approach were suitable.

The author gives an excellent discussion of current practices in Europe with historical context, giving credence to the need for this type of technology. 

Discussion:

  • If possible, the understanding of some of the Tables would be easier if they were in graphical form. Specifically, a graph showing the release rates during the 4 different release zones (Table 2). This sort of release mechanism is widely used in drug release of hydrogels. The author may benefit from fitting a sigmoidal or appropriate release kinetics equation to their system. Two articles that may be of use:
    • Lee, P. I. (1985). Kinetics of drug release from hydrogel matrices. Journal of Controlled Release2, 277-288.
    • Li, J., & Mooney, D. J. (2016). Designing hydrogels for controlled drug delivery. Nature reviews. Materials1(12), 16071. https://doi.org/10.1038/natrevmats.2016.71

Taking into consideration this observation, we developed the section 3.2. Biodegradability of Fertilizers in Aqueous or Composting Environments, by adding figures 8 and 9 and their regression equations in the new table 4.

With special reference to the section 3.3. Leaching the oxidable compounds in laboratory dynamic systems, the previous Table 2 has been removed and replaced by the Fig. 10 and the sigmoidal form has been obtained according to the references recommended to be cited by the reviewer. Moreover, these experimental data have been fitted according to Rigger-Pappas equation, while the corresponding kinetic parameters are presented in the new table 6 and commented in the text. Moreover, a detail kinetic analysis on transport mechanism has been carried out by calculation of kinetic parameters (a new table, nr.7) based on logarithmic form of equation, and the two linear regions have been adequately interpreted and discussed.

Round 2

Reviewer 2 Report

Sorry but even if the changes have significantly improved the manuscript, I strongly doubt of the reproducibility of the experiments. The materials are not evaluate and characterize deeply enough for their publication on Materials. In my opinion this manuscript falls out of the scope.

Author Response

Sorry but even if the changes have significantly improved the manuscript, I strongly doubt of the reproducibility of the experiments. The materials are not evaluate and characterize deeply enough for their publication on Materials. In my opinion this manuscript falls out of the scope.

As referring to the reproducibility of the experiments there is no doubt, as three experimental variants are reported at laboratory scale, while the fourth is at pilot plant. Moreover, as one can see in the technology scheme from figure 3 for obtaining smart agro-collagen hydrogels, for all the reagents there are intervals of concentrations, therefore the reproducibility of these materials is not a real problem in the case studied.

As for the characterization of novel materials, several new tables and figures have been added for physical-chemical analyses of both raw hide waste and the materials obtained, with focus on the tests devoted for applications as fertilisers: biodegradability and oxygen compound release.

When discuss whether this manuscript falls out of the Materials journal scope, one can remind the definition of material: “anything that serves as crude or raw matter to be used or developed”.

In our approach we applied the innovative concept of circular economy by capitalizing a raw material - leather waste dangerous for environment and developing a new process for obtaining useful materials – smart fertilisers for poor soil amendment.

Moreover, here it is the scope announced by Materials journal: “Materials provides a forum for publishing papers which advance the in-depth understanding of the relationship between the structure, the properties or the functions of all kinds of materials. Chemical syntheses, chemical structures and mechanical, chemical, electronic, magnetic and optical properties and various applications will be considered.”

            One may reasonable conclude that this article falls in a high extent into the journal scope.